# The Effect of Carbohydrate Restriction on Lipids, Lipoproteins, and Nuclear Magnetic Resonance-Based Metabolites: CALIBER, a Randomised Parallel Trial

**DOI:** 10.3390/nu15133002

**Published:** 2023-06-30

**Authors:** D. McCullough, T. Harrison, K. J. Enright, F. Amirabdollahian, M. Mazidi, K. E. Lane, C. E. Stewart, I. G. Davies

**Affiliations:** 1Carnegie School of Sport, Leeds Beckett University, Leeds LS6 3QS, UK; d.mccullough@leedsbeckett.ac.uk; 2Research Institute of Sport and Exercise Science, Liverpool John Moores University, Liverpool L3 3AF, UK; k.j.enright@ljmu.ac.uk (K.J.E.); k.e.lane@ljmu.ac.uk (K.E.L.); c.e.stewart@ljmu.ac.uk (C.E.S.); 3Department of Clinical Sciences and Nutrition, University of Chester, Chester CH1 4BJ, UK; t.harrison@chester.ac.uk; 4School of Health and Society, University of Wolverhampton, Wolverhampton WV1 1LY, UK; f.amirabdollahian@wlv.ac.uk; 5Medical Research Council Population Health Research Unit, University of Oxford, Oxford OX1 7LF, UK; mohsen.mazidi@ndph.ox.ac.uk; 6Clinical Trial Service Unit and Epidemiological Studies Unit (CTSU), Nuffield Department of Population Health, University of Oxford, Oxford OX3 7LF, UK; 7Department of Twin Research & Genetic Epidemiology, South Wing St Thomas’, King’s College London, London SE1 7EH, UK

**Keywords:** lipoprotein, cardiovascular disease risk, diet, carbohydrate, fat, NMR spectroscopy

## Abstract

Low-carbohydrate high-fat (LCHF) diets can be just as effective as high-carbohydrate, lower-fat (HCLF) diets for improving cardiovascular disease risk markers. Few studies have compared the effects of the UK HCLF dietary guidelines with an LCHF diet on lipids and lipoprotein metabolism using high-throughput NMR spectroscopy. This study aimed to explore the effect of an ad libitum 8-week LCHF diet compared to an HCLF diet on lipids and lipoprotein metabolism and CVD risk factors. For 8 weeks, *n* = 16 adults were randomly assigned to follow either an LCHF (*n* = 8, <50 g CHO p/day) or an HCLF diet (*n* = 8). Fasted blood samples at weeks 0, 4, and 8 were collected and analysed for lipids, lipoprotein subclasses, and energy-related metabolism markers via NMR spectroscopy. The LCHF diet increased (*p* < 0.05) very small VLDL, IDL, and large HDL cholesterol levels, whereas the HCLF diet increased (*p* < 0.05) IDL and large LDL cholesterol levels. Following the LCHF diet alone, triglycerides in VLDL and HDL lipoproteins significantly (*p* < 0.05) decreased, and HDL phospholipids significantly (*p* < 0.05) increased. Furthermore, the LCHF diet significantly (*p* < 0.05) increased the large and small HDL particle concentrations compared to the HCLF diet. In conclusion, the LCHF diet may reduce CVD risk factors by reducing triglyceride-rich lipoproteins and improving HDL functionality.

## 1. Introduction

Cardiovascular diseases (CVDs) are the leading cause of death, accounting for approximately 17.8 M deaths globally [1]. In 2019, an estimated 12.7 M new cases of CVD were reported, leading to approximately 113 M people living with CVD across European nations alone [2]. To combat CVDs, global bodies recommend limiting dietary fat (particularly saturated fatty acids (SFA)) but encouraging a high dietary intake of carbohydrates (>50% of energy intake, although low in sugar) [3,4]. However, low-carbohydrate (<26% of energy intake) high-fat (LCHF) diets haveshown to perform at least as well as higher-carbohydrate, lower-fat (HCLF) diets in reducing body fat and improving metabolic profiles even with an increase in SFA intake [5]. 

One concern with LCHF diets is an increase in low-density lipoprotein cholesterol (LDL-C) compared to HCLF diets [6]. Interestingly, after 12 months, these differences disappeared, which may, however, be a dietary compliance issue [7]. While LDL-C is a risk marker for CVD, it does not discriminate between LDL size and particle number, with small dense LDL (sdLDL) and higher LDL particle numbers being stronger predictors of CVD risk [8,9]. Therefore, it is necessary to distinguish between them to estimate CVD risk. Apolipoprotein B (ApoB), the primary apolipoprotein of LDL, is directly proportional to LDL particle number and is also superior to LDL-C at determining CVD risk [10]. Concerning this, two main phenotypes (A and B) have been described; phenotype A is characterised by the prevalence of large buoyant LDL, whereas phenotype B is characterised by the prevalence of small dense LDL, and the latter is strongly associated with metabolic disease [9,11,12]. Not only are lipoprotein concentrations a key risk factor for CVD but metabolites such as circulating branched-chain amino acids (BCAAs) are also associated with biomarkers of metabolic diseases and CVD risk [13,14]. As diet exerts a myriad of effects on global metabolism [15], the measurement of all lipoproteins with amino acid metabolites may enable a clearer understanding of the association between CVD risk and LCHF diets. 

Nuclear magnetic resonance (NMR) spectroscopy is a powerful tool that can accurately quantify lipoprotein subclasses (density, size, and particle number) and their associated lipid species [16]. Higher very-low-density lipoprotein (VLDL) and LDL particle concentrations are associated with an elevated CVD risk [17,18]. In contrast, lipid concentrations within large and medium high-density lipoprotein (HDL) particles are inversely associated with CVD risk [17,18]. Large cohort studies have also demonstrated that healthy eating patterns, characterised by high levels of polyunsaturated fatty acids (PUFA) are associated with lower VLDL and LDL-associated lipids, resulting in lower CVD risk [19]. Similarly, controlled studies have highlighted that replacing SFA with PUFA lowers atherogenic particles of the lipoprotein subclasses of VLDL, intermediate-density lipoprotein (IDL), and LDL [20]. However, an LCHF (<20% carbohydrate) diet with higher levels of both dietary SFA and PUFA still improves the lipoprotein particle concentration in comparison to a high-carbohydrate diet, with no difference between LDL-C [21]. Additionally, the consumption of fatty fish is associated with lower levels of tyrosine and valine levels, indicating that dietary fat intake may also modulate amino acid metabolism and may be associated with altered CVD risk [22]. These results indicate that, in contrast to worldwide recommendations [4], reducing dietary carbohydrates rather than SFA may exert greater reductions in the atherogenic lipoprotein profile and improve CVD risk factors. Specifically, the effect of the current UK dietary guidelines [3] in comparison with an LCHF diet on the lipoprotein profile and CVD risk factors has not been explored using NMR spectroscopy.

Therefore, this study aimed to investigate the impact of an ad libitum 8-week LCHF diet compared to an HCLF diet (current UK guidelines) [3] on the global lipid and amino acid profiles in adults with an elevated metabolic risk. We hypothesise that the LCHF diet modifies lipids and lipoprotein profiles that benefit metabolic health, thereby lowering CVD risk.

## 2. Materials and Methods

All procedures followed the CONSORT guidelines for reporting randomised trials [23,24].

### 2.1. Study Design and Recruitment

Details of the study design were reported previously [24] with ethical approval from the Liverpool John Moores University research ethics committee (REC number: 16/ELS/029). This study is registered as a clinical trial (REF: NCT03257085). Participants were included if they were aged 19–64 years with a BMI of 18.5–29.9 kg/m^2^ and excluded if they were a smoker, vegan/vegetarian, took dietary supplements, had any known food allergies or intolerances, consumed alcohol above the weekly UK recommendations, were pregnant, suffered from an eating disorder, suffered from current or previous renal impairment, had a history of cardiometabolic diseases or took lipid, blood pressure or blood glucose-lowering medication. Briefly, after the screening, all participants provided written informed consent and were randomly assigned to either an ad libitum HCLF (*n* = 8) or an LCHF diet (*n* = 8) for 8 weeks using a computerised random allocation sequence and concealed in envelopes. Participants in the HCLF group were required to consume a diet composed of 50% carbohydrate, 15% protein, and at most 35% fat per day (based on the UK Eatwell Guide) [3]. The LCHF group was instructed to consume a diet consisting of ≤50 g of carbohydrates per day to induce ketosis [25] and increase the amount of fat consumed while consuming similar amounts of protein compared to the HCLF group. At 0, 4, and 8 weeks of the diet, fasting whole blood was collected by trained phlebotomists from the antecubital fossa vein and centrifuged at 3000× *g* for 15 min at 4 °C to harvest plasma and serum, which were stored at −80 °C until analysis. Anthropometrics, body composition, blood pressure, physical activity, and dietary intake were recorded at each time point.

### 2.2. NMR Spectroscopy

To quantify lipid, lipoprotein subclasses, fatty acid composition, amino acids, ketone bodies, glycolysis, and Krebs cycle-related metabolites from overnight fasting EDTA plasma, we used the service provided by Nightingale Health Ltd., Helsinki, Finland, which employs a high-throughput proton NMR spectroscopy platform. Lipoprotein subclasses were quantified using lipid concentrations within fourteen subclasses, abundant proteins, and various low-molecular-weight metabolites. The applications and experimentation details of the NMR metabolomics platform were described previously [26,27]. The quantified biomarker measures rather than the NMR spectral data were analysed in relation to clinical/risk factor variables in this study, and examples of spectral annotation were published previously [26,28]. Biomarker quantification was performed in regions where EDTA signals do not overlap, and NMR-based quantification reported comparable results to routine lipid measures and fatty acid measures from gas chromatography [27]. Representative coefficients of variations for the metabolic biomarkers were published previously [29], and all metabolites fell within the range of detection.

### 2.3. Statistics

All normally distributed data are presented as mean ± SD, whereas non-normally distributed data are presented as median ± interquartile range (IQR). All data were explored for distribution using the Shapiro–Wilks test. Normally distributed data underwent a 2 × 3 mixed ANOVA with 2 between factors (LCHF vs. HCLF) and 3 within factors (baseline vs. interim vs. endpoint) to investigate significant differences for the main and interaction effects. If repeated measures data had a missing value, mixed effects analysis was used instead of ANOVA. Non-normally distributed data were log or square root transformed prior to parametric or non-parametric analyses (Friedman and Kruskal–Wallis test). All *p*-values were corrected for multiple testing using the Benjamini and Hochberg method [30] and considered significant at *p* < 0.05. The fold-change percentage from baseline to 8 weeks of the diet was calculated as 100 × (mean C − mean A)/mean A. The GraphPad Prism (San Diego, CA, USA) statistical software was used for statistical analysis. Random forest with a combination of unbiased variable selection framework and repeated double cross-validation was applied using R, version 4.0.3 (R Foundation for Statistical Computing, Vienna, Austria), to detect a panel of metabolites representative of the LCHF diet relative to the HCLF diet at the end of the study (week 8). This method fits many classification trees to a data set and then combines the predictions from all trees to present a final predictive model that ranks variables based on their predictive power. The model underwent extensive tuning to optimise its hyperparameters and mitigate overfitting. As a result, it achieved performance metrics, with an R2 value of 0.62 and a Q2 value of 0.64 (an estimate of the predictive ability of the model calculated by cross-validation). Model performance was confirmed via permutation analysis (*n* = 1000). 

## 3. Results

Participant recruitment, changes in metabolic markers, body composition, and dietary intake were reported previously [24]. The participants consisted of four males and females in the LCHF group (*n* = 8) and five males and three females in the HCLF group (*n* = 8). Participants’ mean age was similar between the groups (LCHF, 43.8 ± 10.4; HCLF, 44.6 ± 15.27; *p* = 0.895). Briefly, no change in dietary intake was reported in the HCLF group during the intervention; however, as reported previously, the percentage of energy derived from fat increased from 34 ± 4 to 61 ± 6%, carbohydrate decreased from 42 ± 9 to 10 ± 4% (both *p* < 0.001) in the LCHF group, and total energy intake was similar between the groups [24]. Body mass decreased (−3.14 kg) in the LCHF group, but remained unchanged in the HCLF group during the intervention [24].

### 3.1. Lipid and Lipoprotein Metabolism

Total cholesterol increased from baseline to 8 weeks following the LCHF (5.09 ± 0.76 to 5.50 ± 0.57 mmol/L, *p* = 0.022) and HCLF (4.55 ± 1.00 to 4.97 ± 1.17 mmol/L, *p* = 0.032) diets, with differential effects on lipoprotein subclass cholesterol concentrations (Figure 1). The changes in cholesterol concentrations were driven by variations in free and esterified cholesterol levels (Table 1). Triglycerides in very large, large, and medium VLDL were significantly (all *p* < 0.05) lower in the LCHF group compared to the HCLF diet group throughout the intervention; however, these did not pass multiple testing comparisons (Table 1). Compared to baseline, 4 weeks of the LCHF diet also resulted in a decrease in triglycerides in the medium (0.05 ± 0.02 to 0.03 ± 0.01 mmol/L, *p* = 0.030) and small (0.05 ± 0.02 to 0.04 ± 0.01 mmol/L, *p* = 0.029) HDL, but returned to baseline levels by week 8.

Both diets also exerted differential responses in lipoprotein phospholipid content (Table 1). At 4 weeks, the level of small LDL phospholipids increased following the HCLF (*p* = 0.023) and LCHF (*p* = 0.013) diets, but IDL phospholipids increased only following the LCHF diet (*p* = 0.032), and large LDL phospholipids increased following the HCLF (*p* = 0.021) diet. Similarly, at 8 weeks, the LCHF diet increased IDL (*p* = 0.010) phospholipids, whereas the HCLF diet increased large (*p* = 0.026) and small LDL (*p* = 0.017) phospholipids. Phospholipids in very large HDL increased (*p* = 0.005) with the LCHF diet compared to the HCLF diet but were not confirmed when corrected for multiple comparisons. Furthermore, sphingomyelins significantly increased from baseline to week 4 (0.48 ± 0.04 to 0.52 ± 0.04 mmol/L, *p* = 0.004) in the LCHF diet only (Appendix A). 

ApoB concentrations increased after 4 (*p* = 0.002) and 8 (*p* = 0.001) weeks of the HCLF diet, and this increase was significantly greater compared to the LCHF diet (*p* = 0.0001) (Figure 2). The HCLF diet also increased small HDL particles after 4 (*p* = 0.019) and 8 (*p* = 0.025) weeks (Appendix A). In contrast, the LCHF diet resulted in increased very small VLDL (*p* = 0.015) particle concentrations (Appendix A). In comparison to HCLF, the LCHF diet resulted in decreased large VLDL (*p* = 0.050) particles and increased very large (*p* = 0.031) and large (*p* = 0.043) HDL particle concentrations (Figure 2) and HDL diameter (*p* = 0.047); however, after multiple comparisons, the significance was lost.

### 3.2. Amino Acids, Glycolysis, and Fatty Acid-Related Metabolites

Both the LCHF and HCLF diets resulted in some alterations in energy-related metabolites (Appendix A). Specifically, glutamine decreased (*p* = 0.013) after 4 weeks of the LCHF diet (0.052 ± 0.074 mmol/L) and was significantly (*p* = 0.002) lower compared to HCLF, (0.062 ± 0.067 mmol/L) but returned to baseline by week 8. Histidine also significantly decreased after 8 weeks (0.074 ± 0.007 to 0.067 ± 0.01 mmol/L, *p* = 0.004) of the LCHF diet. Tyrosine significantly declined following the LCHF diet at 4 and 8 weeks compared to baseline (0.068 ± 0.013 to 0.056 ± 0.015, *p* = 0.004 to 0.058 ± 0.011 mmol/L, *p* = 0.004). Total BCAAs were significantly higher in the LCHF diet group compared to the HCLF diet group at week 4 (0.453 ± 0.053 vs. 0.383 ± 0.053 mmol/L, *p* = 0.021) and week 8 (0.465 ± 0.076 vs. 0.384 ± 0.036 mmol/L, *p* = 0.033). This was primarily driven by the significantly higher valine concentrations in the LCHF diet compared to the HCLF diet at week 4 (0.258 ± 0.035 vs. 0.214 ± 0.023 mmol/L, *p* = 0.010) and week 8 (0.259 ± 0.024 vs. 0.217 ± 0.019 mmol/L, *p* = 0.003). Metabolites of fatty acid oxidation significantly increased with the LCHF diet and were elevated compared to the HCLF diet at 4 weeks (citrate, *p* = 0.040; acetoacetate, *p* = 0.009) and 8 weeks (citrate, *p* = 0.001; acetone, *p* < 0.001) with the LCHF diet. The ketone body 3-Hydroxybutyrate significantly increased after 4 weeks (median (IQR)*:* 0.07 (0.02) to 0.32 (0.42) mmol/L, *p* = 0.008) with the LCHF diet and was significantly elevated compared to HCLF at 4 weeks (median (IQR)*:* 0.32 (0.42) vs. 0.07 (0.07) mmol/L, *p* = 0.007) and 8 weeks (median (IQR)*:* 0.13 (0.18) vs. 0.06 (0.04) mmol/L, *p* = 0.042). Similarly, albumin also increased from baseline to 4 and 8 weeks (42.11 ± 2.68 to 43.44 ± 2.82, *p* = 0.022 to 44.05 ± 1.93 mmol/L, *p* = 0.038) in the LCHF diet group. Furthermore, docosahexaenoic acid significantly increased from baseline to week 4 (0.18 ± 0.03 to 0.25 ± 0.02 mmol/L, *p* < 0.001) in the LCHF diet group only (Appendix A).

### 3.3. Unbiased Random Forest Analysis

Unbiased random forest analysis revealed 12 metabolites related to lipid, lipoprotein, and amino acid metabolism, thereby distinguishing the LCHF diet from HCLF (Table 2).

## 4. Discussion

Unexpectedly, these pilot data show that both an LCHF and HCLF diet resulted in increased lipoprotein cholesterol, which may indicate an increase in CVD risk. However, the LCHF diet resulted in reduced triglyceride-rich lipoproteins and increased HDL phospholipids and particle numbers, indicating an improvement in HDL functionality. These differential effects on lipid and lipoprotein metabolism in a small cohort (*n* = 8 per group) provide some insights into how dietary carbohydrate and fat manipulation may affect CVD risk factors in the short term.

Cholesterol concentrations have long been established as key indicators of CVD risk [31], and diet has been shown as an important regulator [32]. In the current study, participants following either an LCHF or an HCLF diet had increased cholesterol levels, which were due to subtle differences in lipoprotein subclass cholesterol concentrations. The LCHF diet increased XS-VLDL-C, IDL-C, and L-HDL-C, whereas the HCLF diet increased only IDL-C and L-LDL-C. These results are perhaps in contrast with previous research that highlights that although cholesterol concentrations do increase with an LCHF diet, they tend to be greater in LDL-C compared to HCLF [5,7]. However, not all studies show an increase in LDL-C with either diet, even in the absence of decreased body mass [33]. Typically, LDL-C is considered to have a causal effect on CVD [34]; however, a recent Mendelian randomisation (MR) study highlighted that elevated VLDL-C and IDL-C are indicative of increased CVD risk independent of LDL-C [35]. Although HDL-C is associated with lower CVD risk [31], there has been much debate on its role, as elevated HDL-C levels do not appear to protect against CVD in MR analysis [36]. However, more recently, an MR study indicated a protective effect of HDL-C on CVD risk, perhaps due to a larger sample size (*n* = 60,000 vs. *n* = 12,000) compared to Voight et al. (2012) [35]. Therefore, the increase in L-HDL-C with the LCHF diet may be protective against lipoprotein cholesterol increases, whereas this may not occur with the HCLF diet.

The LCHF diet resulted in significantly lower triglycerides in the fractions of VLDL compared to the HCLF diet and significantly decreased triglycerides within HDL. A reduction in triglyceride-rich lipoproteins (TRLs) is shown to be associated with reduced CVD risk [37,38]. TRLs can enter the arterial intima similar to LDL, but their larger size may lead to greater preferential retention of cholesterol-enriched remnants [38,39]. TRLs can also undergo direct phagocytosis, leading to foam cell generation, inflammation, and atherosclerotic plaque formation [38,39]. This reduction in TRLs likely contributes to the reduction in HDL triglycerides via reduced cholesteryl ester transfer protein activity [40]. Elevated HDL triglycerides have been shown to be positively associated with the markers of metabolic disease [40], myocardial infarction, and ischaemic stroke, [37,41] indicating a positive effect of an LCHF diet.

The lipoprotein surface is encompassed by phospholipids, primarily phosphatidylcholine and sphingomyelin, and plays a role in lipoprotein functionality. Sphingomyelin is the precursor to ceramides and sphingomyelins significantly increased following 4 weeks of the LCHF diet. However, it is unclear what effect this may have on CVD risk, as an increase in short-chain sphingomyelins and ceramides are associated with increased CVD risk, whereas longer chains are associated with reduced risk [42]. A decrease in sphingomyelin, particularly in HDL, may also be associated with reduced CVD risk; however, this may be confounded by low HDL-C levels [43]. Interestingly, the phospholipids within very large HDL were significantly elevated in the LCHF diet compared to the HCLF diet, again indicating a reduction in CVD risk, perhaps due to improved HDL functionality [43,44]. The process of cholesterol efflux or HDL functionality has been shown to be associated with reduced CVD risk, independent of HDL-C concentrations [45]. The HCLF diet resulted in increased LDL phospholipid content, which, in contrast, has been shown to decrease after following 8 weeks of the cardioprotective Mediterranean diet [46] where increased MUFA was considered responsible [47].

Lipoprotein particle number and size are strong independent risk factors for the development of CVDs [48]. Restricting dietary carbohydrate intake in adults without disease has shown an increase in LDL particle peak size and reduced number, thereby shifting to a lower-risk phenotype (A) [49]. Although there were large differences in carbohydrate intake in the LCHF groups, meta-regression analysis revealed that this was not a factor; however, weight loss may have influenced the findings [49]. Similar to Falkenhain et al. [49], the current results support the hypothesis that carbohydrate restriction improves LDL particle concentrations. The LCHF diet also decreased large VLDL and increased very large and large HDL particle concentrations relative to the HCLF diet, which is consistent with previous research [33,50]. Although HDL particle concentrations still significantly increased following the HCLF diet, this was accompanied by a significant increase in apolipoprotein B levels, which is used as a surrogate for LDL particle number and is positively associated with CVDs [48]. However, the ratio of apolipoprotein B/A1 or total LDL particle number did not change, indicating that perhaps the small changes in apolipoprotein B are of little clinical significance. The improvements in HDL particle concentrations, along with elevated HDL cholesterol, phospholipids, and reduced TRLs following the LCHF diet, highlight how an LCHF diet may reduce CVD risk by improving HDL functionality and metabolism (Figure 3).

Random forest statistical analysis allowed for the unbiased detection of the effects of both diets on multiple metabolites (Table 2). These data also corroborate primary analyses that the LCHF diet results in greater reductions in triglycerides and increased lipids (driven by cholesterol and phospholipids) in HDL compared to the HCLF diet. Additionally, the LCHF diet appears to increase BCAAs and reduce tyrosine and histidine concentrations relative to the HCLF diet. While reductions in tyrosine and histidine may suggest a reduced CVD risk [51], elevated BCAAs have been associated with CVD and type 2 diabetes [41,52]. Increases in BCAAs could be due to increased protein intake and the release of amino acids for gluconeogenesis as a normal part of LCHF adaptations; however, long-term controlled studies are required on these implications.

Like all research, the current study has its limitations, which were previously highlighted [24]. Briefly, the intervention was short, with a small sample size in individuals without CVD; therefore, the results should not be extrapolated to long-term health or generalised to populations with CVD. Although dietary records show good adherence to the LCHF diet, this may not be the case, as ketone markers were highest by week 4 (but below the ketosis threshold of 0.5 mmol/L) and decreased by week 8, which may have reduced the impact of the LCHF diet on biomarkers of CVD risk at this stage. Nonetheless, the objective of this study was to investigate changes in lipid and lipoprotein metabolism, rather than long-term compliance, and these changes were evident. The strengths of this study include the use of high-throughput NMR spectroscopy to identify discrete lipoprotein subclasses that can infer CVD risk and elucidate the regulatory role of diet.

In conclusion, following a short-term LCHF diet may reduce TRLs and improve HDL metabolism and functionality. The potential improvements in HDL functionality may compensate for the increases in VLDL/IDL/LDL cholesterol, but this may not be apparent with an HCLF diet. It is unclear how this may be used to infer the overall CVD risk due to the small sample size and short duration of the study. Furthermore, the LCHF diet increases BCAA levels, which may be associated with an elevated CVD risk; however, this could also be reflective of amino acid release for gluconeogenesis. Long-term studies with large cohorts are warranted to confirm the role of increased dietary fat and carbohydrate restriction in lipoprotein metabolism and CVD risk factors.

## Figures and Tables

**Figure 1 nutrients-15-03002-f001:**
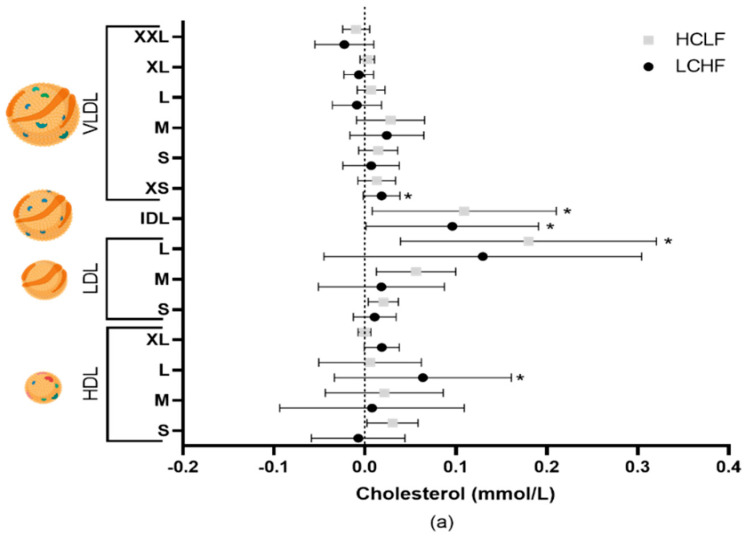
The effects of LCHF (*n* = 8) and HCLF diets (*n* = 8) for 8 weeks on lipoprotein subclass cholesterol concentration. (**a**) The mean difference ± SD in lipoprotein cholesterol from baseline to 4 weeks following LCHF and HCLF. (**b**) The mean difference ± SD in lipoprotein cholesterol from baseline to 8 weeks following LCHF and HCLF. Lipoprotein figures were created using BioRender.com. HCLF, high-carbohydrate low-fat diet; LCHF, low-carbohydrate high-fat diet; * *p* < 0.05, denotes the significant effect of time; ** *p* < 0.01, denotes the significant effect of time, ^#^ *p* < 0.05, denotes significant differences between diets.

**Figure 2 nutrients-15-03002-f002:**
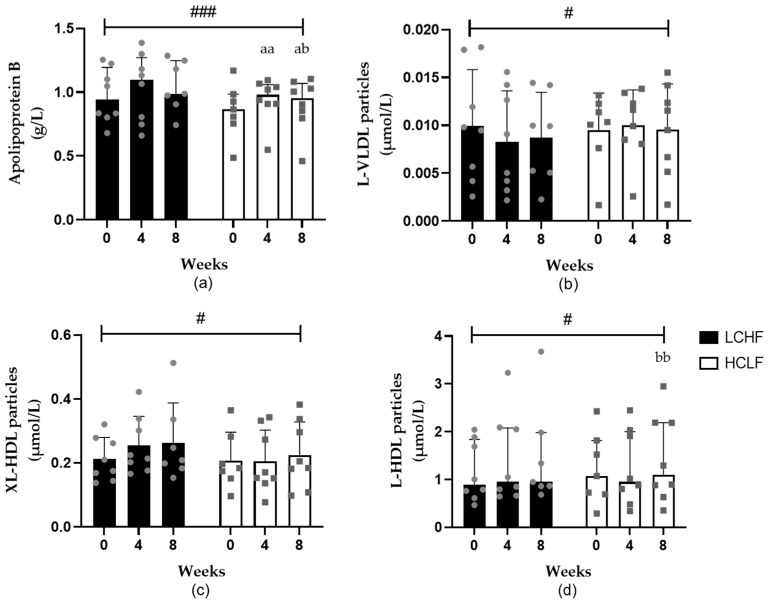
The effects of LCHF (*n* = 8) and HCLF diets (*n* = 8) for 8 weeks on lipoprotein particle concentration. (**a**) The effect (median ± IQR) of LCHF and HCLF diets on fasting apolipoprotein B concentrations. (**b**) The effect (mean ± SD) of LCHF and HCLF diets on fasting L-VLDL particle concentrations. (**c**) The effect (mean ± SD) of LCHF and HCLF diets on fasting XL-HDL particle concentrations. (**d**) The effect (median ± IQR) of LCHF and HCLF diets on fasting L-HDL particle concentrations. The grey symbols represent individual responses to the LCHF diet (circle) and HCLF diet (square). HCLF, high-carbohydrate low-fat diet; LCHF, low-carbohydrate high-fat diet. ^#^ *p* < 0.05 and ^###^ *p* < 0.001 denote significant interactions between groups; ^a^ *p* < 0.05, ^aa^ *p* < 0.01 denotes significant difference to week 0; ^b^ *p* < 0.05 and ^bb^ *p* < 0.01 denote significant difference in week 4.

**Figure 3 nutrients-15-03002-f003:**
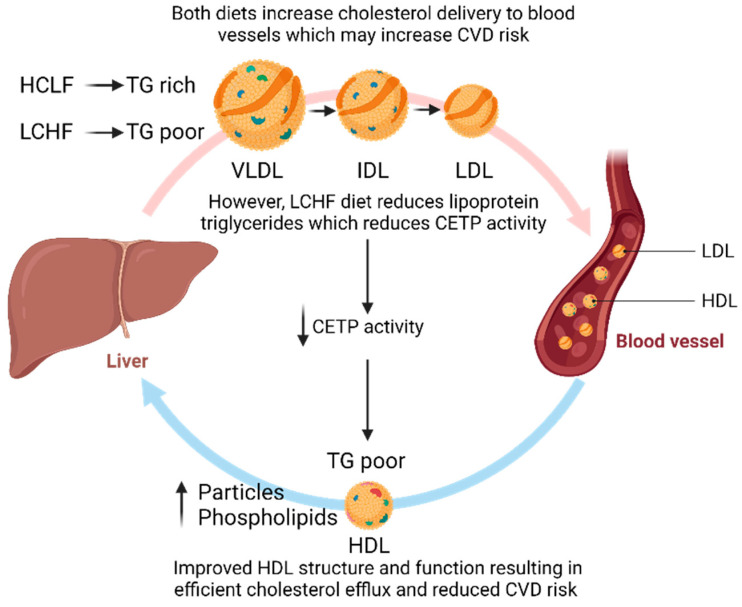
The potential effect of an LCHF diet on improving HDL functionality to reduce CVD risk. CETP, cholesterol ester transfer protein; HDL, high-density lipoprotein; IDL, intermediate-density lipoprotein; LDL, low-density lipoprotein; TG, triglycerides; VLDL, very low-density lipoprotein. Figures were created using BioRender.com.

**Table 1 nutrients-15-03002-t001:** The effect of LCHF (*n* = 8) and HCLF (*n* = 8) diets on lipids in lipoprotein subclasses.

	LCHF					HCLF			*p* Value		
	Week 0	Week 4	Week 8	95% CI Difference 0–8 Weeks	Week 0	Week 4	Week 8	95% CI Difference 0–8 Weeks	Group	Time	Time × Diet
**Total Lipids**											
Total cholesterol (mmol/L)	5.09 ± 0.76	5.44 ± 0.86	5.50 ± 0.57 ^a^	0.08, 0.73	4.55 ± 1.00	5.03 ± 0.95 ^a^	4.97 ± 1.17 ^a^	0.05, 0.79	0.393	**0.010**	0.967
Non-HDL-C (mmol/L)	3.84 ± 0.85	4.11 ± 1.06	4.15 ± 0.76	−0.01, 0.64	3.30 ± 0.94	3.72 ± 0.82 ^a^	3.58 ± 0.97	−0.01, 0.58	0.364	**0.013**	0.951
Remnant cholesterol (mmol/L)	1.75 ± 0.44	1.86 ± 0.55	1.95 ± 0.39	−0.03, 0.37	1.52 ± 0.44	1.69 ± 0.36	1.59 ± 0.42	−0.12, 0.26	0.342	0.062	0.605
Total esterified cholesterol (mmol/L)	3.71 ± 0.53	3.96 ± 0.59	3.99 ± 0.39 ^a^	0.05, 0.52	3.31 ± 0.72	3.67 ± 0.69 ^a^	3.63 ± 0.86 ^a^	0.06, 0.59	0.399	**0.009**	0.978
Total free cholesterol (mmol/L)	1.39 ± 0.23	1.48 ± 0.28	1.51 ± 0.18	0.03, 0.22	1.24 ± 0.28	1.36 ± 0.26	1.34 ± 0.31	−0.02, 0.20	0.383	**0.017**	0.855
Total triglycerides (mmol/L)	1.19 ± 0.59	0.99 ± 0.43	1.06 ± 0.34	−0.42, 0.15	1.15 ± 0.38	1.15 ± 0.31	1.13 ± 0.41	−0.14, 0.11	0.695	0.171	0.095
Total lipids in lipoprotein particles (mmol/L)	9.23 ± 1.53	9.45 ± 1.51	9.66 ± 1.04	−0.15, 1.01	8.47 ± 1.47	9.1 ± 1.59	9.05 ± 1.87	−0.06, 1.23	0.607	0.189	0.855
**VLDL lipids**											
VLDL cholesterol (mmol/L)	0.79 (0.42)	0.86 (0.59)	0.84 (0.41)	−0.05, 0.19	0.72 (0.18)	0.83 (0.14)	0.74 (0.23)	−0.12, 0.11	0.381	0.278	0.400
Cholesteryl esters in VLDL (mmol/L)	0.48 (0.25)	0.54 (0.34)	0.51 (0.24)	−0.02, 0.13	0.43 (0.11)	0.51 (0.08)	0.46 (0.12)	−0.07, 0.07	0.616	0.364	0.281
Free cholesterol in VLDL (mmol/L)	0.31 ± 0.13	0.31 ± 0.14	0.33 ± 0.11	−0.03, 0.07	0.28 ± 0.10	0.30 ± 0.09	0.28 ± 0.10	−0.05, 0.04	0.726	0.584	0.494
Triglycerides in VLDL (mmol/L)	0.82 ± 0.48	0.65 ± 0.37	0.69 ± 0.33	−0.36, 0.09	0.79 ± 0.29	0.80 ± 0.27	0.78 ± 0.35	−0.13, 0.11	0.637	0.207	0.060
Phospholipids in chylomicrons and extremely large VLDL (mmol/L)	0.03 ± 0.04	0.02 ± 0.02	0.03 ± 0.03	−0.02, 0.01	0.03 ± 0.02	0.02 ± 0.02	0.02 ± 0.02	−0.02, 0.00	0.962	**0.043**	0.443
Triglycerides in chylomicrons and extremely large VLDL (mmol/L)	0.07 (0.20)	0.04 (0.12)	0.08 (0.12)	−0.03, 0.00	0.12 (0.05)	0.07 (0.10)	0.07 (0.09)	0.00, 0.03	0.280	0.828	**0.014**
Phospholipids in very large VLDL (mmol/L)	0.04 ± 0.03	0.03 ± 0.02	0.03 ± 0.02	−0.01, 0.01	0.04 ± 0.02	0.04 ± 0.02	0.03 ± 0.02	−0.01, 0.00	0.973	0.248	0.126
Triglycerides in very large VLDL (mmol/L)	0.10 ± 0.08	0.06 ± 0.06	0.07 ± 0.05	−0.07. 0.01	0.09 ± 0.05	0.09 ± 0.04	0.09 ± 0.06	−0.02, 0.01	0.647	0.126	**0.040**
Phospholipids in large VLDL (mmol/L)	0.07 ± 0.04	0.05 ± 0.04	0.06 ± 0.04	−0.03, 0.01	0.06 ± 0.03	0.06 ± 0.03	0.06 ± 0.03	−0.01, 0.01	0.778	0.294	0.053
Triglycerides in large VLDL (mmol/L)	0.15 ± 0.08	0.12 ± 0.08	0.12 ± 0.06	−0.07, 0.02	0.14 ± 0.05	0.15 ± 0.06	0.15 ± 0.07	−0.02, 0.04	0.511	0.438	**0.029**
Phospholipids in medium VLDL (mmol/L)	0.14 (0.08)	0.16 (0.12)	0.15 (0.07)	0.00, 0.01	0.13 (0.04)	0.16 (0.02)	0.14 (0.05)	−0.01, 0.00	0.514	0.709	0.063
Triglycerides in medium VLDL (mmol/L)	0.26 ± 0.11	0.23 ± 0.11	0.22 ± 0.09	−0.10, 0.03	0.25 ± 0.08	0.27 ± 0.08	0.27 ± 0.10	−0.02, 0.05	0.474	0.465	**0.027**
Phospholipids in small VLDL (mmol/L)	0.11 (0.05)	0.12 (0.08)	0.11 (0.05)	−0.01, 0.02	0.10 (0.03)	0.12 (0.02)	0.11 (0.03)	−0.01, 0.02	0.489	0.914	0.529
Triglycerides in small VLDL (mmol/L)	0.14 ± 0.07	0.12 ± 0.05	0.12 ± 0.04	−0.05, 0.02	0.15 ± 0.05	0.14 ± 0.03	0.14 ± 0.05	−0.02, 0.01	0.418	0.198	0.166
Phospholipids in very small VLDL (mmol/L)	0.11 ± 0.03	0.11 ± 0.03	0.12 ± 0.02	0.00, 0.03	0.10 ± 0.02	0.10 ± 0.02	0.10 ± 0.02	−0.02, 0.01	0.345	0.482	0.132
Triglycerides in very small VLDL (mmol/L)	0.06 ± 0.02	0.06 ± 0.02	0.06 ± 0.01	−0.05, 0.02	0.06 ± 0.02	0.06 ± 0.01	0.06 ± 0.02	−0.02, 0.01	0.914	0.266	0.917
**IDL lipids**											
Phospholipids in IDL (mmol/L)	0.33 ± 0.06	0.36 ± 0.08 ^a^	0.37 ± 0.05 ^a^	0.01, 0.07	0.28 ± 0.07	0.31 ± 0.06	0.30 ± 0.07	−0.01, 0.05	0.188	**0.020**	0.400
Triglycerides in IDL (mmol/L)	0.10 ± 0.03	0.09 ± 0.02	0.10 ± 0.01	−0.02, 0.03	0.09 ± 0.02	0.09 ± 0.01	0.09 ± 0.02	−0.01, 0.01	0.570	0.664	0.973
**LDL lipids**											
Clinical LDL cholesterol (mmol/L)	3.16 ± 0.72	3.45 ± 0.88	3.43 ± 0.66	−0.03, 0.58	2.63 ± 0.85	3.05 ± 0.72 ^a^	2.94 ± 0.88 ^a^	0.06, 0.56	0.308	**0.005**	0.957
LDL cholesterol (mmol/L)	2.09 ± 0.44	2.24 ± 0.52	2.2 ± 0.37	−0.05, 0.29	1.78 ± 0.51	2.03 ± 0.46 ^a^	1.99 ± 0.56 ^a^	0.08, 0.35	0.393	**0.009**	0.662
Cholesteryl esters in LDL (mmol/L)	1.52 ± 0.33	1.61 ± 0.38	1.59 ± 0.28	−0.06, 0.20	1.30 ± 0.37	1.48 ± 0.34 ^a^	1.45 ± 0.41 ^a^	0.06, 0.25	0.455	**0.016**	0.519
Free cholesterol in LDL (mmol/L)	0.57 ± 0.11	0.63 ± 0.13	0.62 ± 0.10	0.00, 0.01	0.48 ± 0.14	0.56 ± 0.12 ^a^	0.54 ± 0.15 ^a^	0.02, 0.10	0.256	**0.003**	0.932
Triglycerides in LDL (mmol/L)	0.15 ± 0.04	0.14 ± 0.03	0.15 ± 0.02	−0.04, 0.04	0.14 ± 0.04	0.14 ± 0.02	0.14 ± 0.03	−0.01, 0.02	0.543	0.731	0.917
Phospholipids in large LDL (mmol/L)	0.42 ± 0.08	0.45 ± 0.10	0.45 ± 0.07	0.00, 0.06	0.36 ± 0.10	0.41 ± 0.08 ^a^	0.40 ± 0.10 ^a^	0.01, 0.06	0.312	**0.007**	0.980
Triglycerides in large LDL (mmol/L)	0.10 ± 0.03	0.10 ± 0.02	0.10 ± 0.01	−0.02, 0.03	0.09 ± 0.02	0.09 ± 0.01	0.09 ± 0.02	−0.01, 0.01	0.407	0.888	0.973
Phospholipids in medium LDL (mmol/L)	0.19 (0.07)	0.21 (0.08)	0.19 (0.05)	−0.01, 0.02	0.17 (0.04)	0.20 (0.03)	0.19 (0.04)	0.01, 0.03	0.833	0.076	0.496
Triglycerides in medium LDL (mmol/L)	0.03 ± 0.01	0.03 ± 0.01	0.03 ± 0.04	−0.01, 0.01	0.03 ± 0.01	0.03 ± 0.01	0.03 ± 0.01	0.00, 0.00	0.699	0.568	0.739
Phospholipids in small LDL (mmol/L)	0.10 ± 0.02	0.11 ± 0.02 ^a^	0.11 ± 0.01	0.00, 0.01	0.09 ± 0.02	0.10 ± 0.01 ^a^	0.10 ± 0.02 ^a^	0.00, 0.00	0.174	**0.003**	0.318
Triglycerides in small LDL (mmol/L)	0.02 ± 0.01	0.01 ± 0.01	0.01 ± 0.01	−0.01, 0.00	0.01 ± 0.01	0.01 ± 0.01	0.01 ± 0.01	0.00, 0.00	0.985	0.199	0.547
**HDL lipids**											
HDL cholesterol (mmol/L)	1.22 (0.28)	1.14 (0.52)	1.19 (0.24)	−0.13, 0.31	1.29 (0.53)	1.27 (0.35)	1.37 (0.52)	−0.03, 0.30	0.936	0.208	0.653
Cholesteryl esters in HDL (mmol/L)	0.95 (0.22)	0.89 (0.40)	0.93 (0.18)	−0.10, 0.23	1.03 (0.41)	1.01 (0.27)	1.09 (0.40)	−0.02, 0.23	0.888	0.196	0.703
Free cholesterol in HDL (mmol/L)	0.27 (0.06)	0.25 (0.11)	0.27 (0.06)	−0.03, 0.09	0.26 (0.10)	0.26 (0.08)	0.28 (0.11)	−0.01, 0.07	0.880	0.266	0.524
Triglycerides in HDL (mmol/L)	0.13 ± 0.05	0.10 ± 0.03	0.12 ± 0.03 ^b^	−0.03, 0.02	0.13 ± 0.04	0.12 ± 0.02	0.12 ± 0.03	−0.03, 0.01	0.470	**0.041**	0.335
Phospholipids in very large HDL (mmol/L)	0.04 (0.06)	0.06 (0.07)	0.05 (0.06)	−0.02, 0.06	0.05 (0.03)	0.04 (0.06)	0.05 (0.06)	−0.02, 0.03	0.579	0.208	**0.005**
Triglycerides in very large HDL (mmol/L)	0.01 ± 0.01	0.01 ± 0.01	0.01 ± 0.01	0.00, 0.00	0.01 ± 0.01	0.01 ± 0.01	0.01 ± 0.01	0.00, 0.00	0.957	0.325	0.881
Phospholipids in large HDL (mmol/L)	0.20 (0.20)	0.20 (0.28)	0.20 (0.18)	−0.07, 0.19	0.23 (0.17)	0.21 (0.23)	0.24 (0.27)	−0.04, 0.12	0.933	0.181	0.148
Triglycerides in large HDL (mmol/L)	0.02 ± 0.01	0.02 ± 0.01	0.03 ± 0.01	−0.01, 0.01	0.02 ± 0.01	0.02 ± 0.01	0.02 ± 0.01	−0.01, 0.01	0.648	0.147	0.556
Phospholipids in medium HDL (mmol/L)	0.42 (0.06)	0.38 (0.13)	0.42 (0.06)	−0.07, 0.07	0.45 (0.08)	0.43 (0.08)	0.47 (0.14)	−0.02, 0.09	0.465	0.369	0.865
Triglycerides in medium HDL (mmol/L)	0.05 ± 0.02	0.03 ± 0.01 ^a^	0.04 ± 0.01 ^bb^	−0.01, 0.01	0.05 ± 0.02	0.04 ± 0.01	0.04 ± 0.01	−0.01, 0.01	0.378	**0.026**	0.215
Phospholipids in small HDL (mmol/L)	0.65 ± 0.06	0.61 ± 0.07	0.63 ± 0.06	−0.06, 0.02	0.64 ± 0.08	0.66 ± 0.09	0.67 ± 0.10	−0.01, 0.07	0.353	0.647	0.133
Triglycerides in small HDL (mmol/L)	0.05 ± 0.02	0.04 ± 0.01 ^a^	0.04 ± 0.01	−0.02, 0.01	0.05 ± 0.02	0.05 ± 0.01	0.05 ± 0.01	−0.01, 0.00	0.554	**0.024**	**0.071**

Values are expressed as mean ± SD or median (interquartile range) of *n* = 8 LCHF and *n* = 8 HCLF. ^a^
*p* < 0.05, denotes significant difference to week 0; ^b^
*p* < 0.05 and ^bb^
*p* < 0.01 denotes significant difference to week 4. Bold text highlights significant results (*p* < 0.05). CI, confidence interval; HCLF, high-carbohydrate low-fat diet; LCHF, low-carbohydrate high-fat diet.

**Table 2 nutrients-15-03002-t002:** Unbiased random forest analysis distinguishing the LCHF diet from the HCLF diet from baseline to 8 weeks.

		Fold Change (%)
Metabolite	LCHF	HCLF
Triglycerides to total lipids ratio in very large HDL	−20.23	−2.07
Isoleucine	16.06	−1.97
Histidine	−10.5	7.25
Leucine	12.7	0.11
Total lipids in large HDL	24.55	14.92
Tyrosine	−13.67	−7.1
The ratio of omega-6 fatty acids to omega-3 fatty acids	−7.73	−1.21
Triglycerides to total lipids ratio in medium LDL	−1.92	−8.08
Cholesterol in small VLDL	10.26	5.01
Total lipids in chylomicrons and extremely large VLDL	−21.01	−25.54
Apolipoprotein B	7.58	5.63
Cholesteryl esters in very large VLDL	−0.53	−2.04

## Data Availability

The data presented in this study are available on request from the corresponding author. The data are not publicly available due to ethical restrictions.

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
