# Peer review of "The Effect of Carbohydrate Restriction on Lipids, Lipoproteins, and Nuclear Magnetic Resonance-Based Metabolites: CALIBER, a Randomised Parallel Trial"

_nutrients, 2023, doi:10.3390/nu15133002_

Round 1

Reviewer 1 Report

In this manuscript, the authors used high-throughput NMR spectroscopy to compare the effects of the UK HCLF dietary guidelines with a LCHF diet on lipid and lipoprotein metabolism. They investigated the effects of LCHF and HCLF on lipid and lipoprotein metabolism and cardiovascular disease risk factors after 8 weeks of free diet. This research is quite meaningful, but the author's research, the description and explanation of the problem is not clear, the conclusion is far-fetched.

1. The individual differences of people are relatively large, so the author only uses 8 people as the research object, is it too little? The metabolome results from two eight-person cohorts are less confident, and the authors do not give confidence intervals for these studies.

2. Please specify the frequency and model of NMR. At the same time, NMR spectra and attribution results should at least be given in the supporting materials.

3. The author claimed that the high-throughput proton NMR spectroscopy platform was used to quantify lipids, but was H spectrum used to detect lipids? If yes, the quantitative results would be inaccurate under the influence of resolution.

4. EDTA plasma will be complexed with macromolecules of blood, which will affect spectral peaks. How does the author consider eliminating it?

5. Figure 2 in line 212 is missing

6. How to calculate the Fold change in Table 2, and the parameters of the classification model of random forest? Is the model overfitted? The other parameters , such as R2 and Q2 (an estimate of the predictive ability of the model calculated by cross-validation) was missing too.

7. The number of references are too much.

Author Response

In this manuscript, the authors used high-throughput NMR spectroscopy to compare the effects of the UK HCLF dietary guidelines with a LCHF diet on lipid and lipoprotein metabolism. They investigated the effects of LCHF and HCLF on lipid and lipoprotein metabolism and cardiovascular disease risk factors after 8 weeks of free diet. This research is quite meaningful, but the author's research, the description and explanation of the problem is not clear, the conclusion is far-fetched.

Comment 1.

The individual differences of people are relatively large, so the author only uses 8 people as the research object, is it too little? The metabolome results from two eight-person cohorts are less confident, and the authors do not give confidence intervals for these studies.

Response:

We agree that the sample size is quite small which reduces the confidence in results. Due to this we applied the Benjamini & Hochberg correction to reduce the false discovery rate or reduce type 1 errors. However, the Benjamini & Hochberg does not provide 95% confidence intervals. We have now calculated the 95% confidence intervals for 0-8 weeks of the diet and included them in the table.

We also agree that the conclusions are difficult to relate to CVD risk due to the small sample size and short duration of the study. We have now adjusted the beginning of the discussion and conclusion to clarify this.

Discussion (Lines 233-239): “Unexpectedly, these pilot data show that both a LCHF and HCLF diet resulted in increased lipoprotein cholesterol which may indicate an increase in CVD risk. However, the LCHF diet resulted in reduced triglyceride rich lipoproteins and increased HDL phospholipids and particle number indicating an improvement in HDL functionality and potential reduced CVD risk (Figure 3). These differential effects on lipid and lipoprotein metabolism in a small cohort (n=8 per group) provides insights on how dietary carbohydrate and fat manipulation may affect CVD risk factors in the short-term.”

Conclusion (322-330): “In conclusion, following a short-term LCHF diet may reduce TRLs and improve HDL metabolism and functionality. The potential improvements in HDL functionality may compensate for increases in VLDL/IDL/LDL cholesterol but this may not be apparent with a HCLF diet. It is unclear how this may infer overall CVD risk due to the small sample size and short duration of the study. Furthermore, the LCHF diet increases BCAA levels which may be associated with elevated CVD risk; however, this could also be reflective of amino acid release for gluconeogenesis. Long-term studies with large cohorts are warranted to confirm the role of increased dietary fat and carbohydrate restriction on lipoprotein metabolism and CVD risk factors.”

Comment 2.

Please specify the frequency and model of NMR. At the same time, NMR spectra and attribution results should at least be given in the supporting materials.

Response to comments 2 – 4

Thank you for this question. We understand the need for such details however, we used Nightingale Health as a service that only provide quantitative results in molarity. The service is validated and been used in over 400 previous publications, but the methods are subject to propriety intellectual rights:

https://research.nightingalehealth.com/blog/next-generation-clinical-chemistry?gclid=EAIaIQobChMI_N28zNrE_wIVh9_tCh02WgNHEAAYAiAAEgIIifD_BwE.

Therefore, we do not receive the detailed spectra nor the precise methods. The service allows the use of EDTA plasma and the principle of the methods cited the following studies in the methods which you can also see in direct responses to each question below:

Soininen et al., (2015) - doi:10.1161/CIRCGENETICS.114.000216.

Julkunen et al., (2023) doi:10.1038/s41467-023-36231-7.

Wurtz et al., (2016) doi:10.1016/j.jacc.2015.12.060.

Wurtz et al. (2017) doi:10.1093/aje/kwx016.

Response 2:

Lines 120-122: The quantified biomarker measures rather than the NMR spectral data were analysed in relation to clinical/risk factor variables in this study and examples of spectral annotation have been published previously [26,28].

Comment 3.

The author claimed that the high-throughput proton NMR spectroscopy platform was used to quantify lipids, but was H spectrum used to detect lipids? If yes, the quantitative results would be inaccurate under the influence of resolution.

Comment 4.

EDTA plasma will be complexed with macromolecules of blood, which will affect spectral peaks. How does the author consider eliminating it?

Response 3 & 4:

Lines 122-125: Biomarker quantification was done in regions where EDTA signals do not overlap and NMR based quantification report comparable results to routine lipid measures and fatty acids measures from gas chromatography [27].

Comment 5.

Figure 2 in line 212 is missing

Response:

Thank you for pointing this out, it must’ve accidentally been removed during editing and has been amended.

Comment 6.

How to calculate the Fold change in Table 2, and the parameters of the classification model of random forest? Is the model overfitted? The other parameters, such as R2 and Q2 (an estimate of the predictive ability of the model calculated by cross-validation) was missing too.

Response:

Thank you for recognising this, the following statements have been added:

Line 134-135 “Fold change percentage of baseline to 8 weeks of diet was calculated as 100 x (mean C – mean A)/mean A.”

Line 157 – 160: The model has undergone extensive tuning to optimize its hyperparameters and mitigate overfitting. As a result, it has achieved performance metrics, with an R2 value of 0.62 and a Q2 value of 0.64

Comment 7.

The number of references are too much.

Response: Thank you for your suggestion, although there doesn’t appear to be a reference limit for this journal, but we agree there are some unnecessary references which have been removed.

Reviewer 2 Report

The article entitled, The Effect Of Carbohydrate Restriction On Lipids, Lipoproteins, And NMR-Based Metabolites: CALIBER, A Randomised Parallel Trial, is a novel study of high interest in this area of nutrition. The introduction and discussion are well-written, with support and rationale for the purpose and results of this study.  The results are laid out clearly.

My comments are for the methods section:

1.      Although another paper cites some of the methods, this paper should still include methods relating to dietary intake adherence and ketone measures – with such a small sample size, it is important to know if the participants adhered to the diet and if they were in nutritional ketosis.

2.      How many males and females were in each group and what was the average age of each group?  Did that differ?

3.      Were medications controlled?

4.      Exclusion/inclusion criteria?

Author Response

The article entitled, The Effect Of Carbohydrate Restriction On Lipids, Lipoproteins, And NMR-Based Metabolites: CALIBER, A Randomised Parallel Trial, is a novel study of high interest in this area of nutrition. The introduction and discussion are well-written, with support and rationale for the purpose and results of this study.  The results are laid out clearly.

My comments are for the methods section:

Comment 1.

Although another paper cites some of the methods, this paper should still include methods relating to dietary intake adherence and ketone measures – with such a small sample size, it is important to know if the participants adhered to the diet and if they were in nutritional ketosis.

Response: Thank you for this comment. We agree and these details of dietary intake are included in results section on lines 151-154 with some extra details to clarify this point which now states: “Briefly, no change in dietary intake was reported in the HCLF group during the intervention; however, as reported previously, the percentage of energy derived from fat increased from 34 ± 4 to 61 ± 6% and carbohydrate decreased from 42 ± 9 to 10 ± 4% (both P < 0.001) in the LCHF group and total energy intake was similar between groups.”

Also, in results section fatty acid metabolites are described and on lines 209-213, the changes in the ketone body 3-Hydroxybutyrate are also described “The ketone body 3-Hydroxybutyrate significantly increased after 4 weeks (median (IQR): 0.07 (0.02) to 0.32 (0.42) mmol/L, P = 0.008) with the LCHF diet and was significantly elevated compared to HCLF at 4 (median (IQR): 0.32 (0.42) vs. 0.07 (0.07) mmol/L, P = 0.007) and 8 weeks (median (IQR): 0.13 (0.18) vs. 0.06 (0.04) mmol/L, P = 0.042).

In the discussion of limitations on lines 314-317, we have discussed the potential reduction in adherence as ketone markers were lower at week 8 vs week 4 but still elevated relative to the HCLF group. “Although dietary records show good adherence to the LCHF diet, this may not be the case as ketone markers were highest by week 4 (but below the nutritional ketosis threshold of 0.5 mmol/L) and decreased by week 8 which may have reduced the impact of the LCHF diet on biomarkers of CVD risk at this stage.”

Comment 2.

How many males and females were in each group and what was the average age of each group?  Did that differ?

Response: Thank you for your comment. The following statement has now been included on lines 148-151, “Participants consisted of 4 males and females in the LCHF group (n=8) and 5 males and 3 females in the HCLF group (n=8). Participants mean age were similar between groups (LCHF, 43.8 ± 10.4; HCLF, 44.6 ± 15.27; P = 0.895).”

Comment 3.

Were medications controlled?

Response: Thank you for this important question. Participants were free from medication as they were excluded if they took any lipid, blood pressure or blood glucose-lowering medication. The addition of exclusion criteria to the manuscript in the below comment should clarify this.

Comment 4.

Exclusion/inclusion criteria?

Response: Thank you for your comment. The following statement has now been included on lines 96-102. “Participants were included if they were aged 19–64 years with a BMI of 18.5–29.9 kg/m2 and excluded if they were a smoker, vegan/vegetarian, took dietary supplements, had any known food allergies or intolerances, consumed alcohol above the weekly UK recommendations, were pregnant, suffered from an eating disorder, suffered from current or previous renal impairment, had a history of cardiometabolic diseases, or took lipid, blood pressure or blood glucose-lowering medication.”

Round 2

Reviewer 1 Report

No comments